# Characterization and Microstructural Evolution of Continuous BN Ceramic Fibers Containing Amorphous Silicon Nitride

**DOI:** 10.3390/ma14206194

**Published:** 2021-10-18

**Authors:** Yang Li, Min Ge, Shouquan Yu, Huifeng Zhang, Chuanbing Huang, Weijia Kong, Zhiguang Wang, Weigang Zhang

**Affiliations:** 1Key Laboratory of Science and Technology on Particle Materials, Key Laboratory of Multiphase Complex Systems, Institute of Process Engineering, Chinese Academy of Sciences, Beijing 100190, China; liyang19@ipe.ac.cn (Y.L.); gemin@ipe.ac.cn (M.G.); sqyu@ipe.ac.cn (S.Y.); hfzhang@ipe.ac.cn (H.Z.); cbhuang@ipe.ac.cn (C.H.); wjkong19@ipe.ac.cn (W.K.); wangzhiguang18@mails.ucas.ac.cn (Z.W.); 2School of Chemical Engineering, University of Chinese Academy of Sciences, Beijing 100049, China; 3School of Rare Earth, University of Science and Technology of China, Ganzhou 341000, China

**Keywords:** composite ceramic fibers, boron nitride, silicon nitride

## Abstract

Boron nitride (BN) ceramic fibers containing amounts of silicon nitride (Si_3_N_4_) were prepared using hybrid precursors of poly(tri(methylamino)borazine) (PBN) and polycarbosilane (PCS) via melt-spinning, curing, decarburization under NH_3_ to 1000 °C and pyrolysis up to 1600 °C under N_2_. The effect of Si_3_N_4_ contents on the microstructure of the BN/Si_3_N_4_ composite ceramics was investigated. Series of the BN/Si_3_N_4_ composite fibers containing various amounts of Si_3_N_4_ from 5 wt% to 25 wt% were fabricated. It was found that the crystallization of Si_3_N_4_ could be totally restrained when its content was below 25 wt% in the BN/Si_3_N_4_ composite ceramics at 1600 °C, and the amorphous BN/Si_3_N_4_ composite ceramic could be obtained with a certain ratio. The mean tensile strength and Young’s modulus of the composite fibers correlated positively with the Si_3_N_4_ mass content, while an obvious BN (shell)/Si_3_N_4_ (core) was formed only when the Si_3_N_4_ content reached 25 wt%.

## 1. Introduction

Advanced antenna radomes or windows are widely used in aerospace radar devices, and wave-transparent materials with high performance are urgently required [1,2,3]. Continuous ceramic fiber reinforced ceramic matrix composites are the most promising materials, and several inorganic continuous fibers, such as quartz, Si_3_N_4_ and BN fibers, are suitable candidates as reinforcements [4,5]. Because of the occurrence of re-crystallization and particle coarsening, quartz fibers undergo serious degradation of mechanical strength at temperatures above 900 °C [6]. Although BN fibers show excellent high-temperature stability, with a melting point above 2900 °C [7,8,9], their low mechanical strength impedes any further development [10,11]. Generally, Si_3_N_4_ fibers possess higher mechanical strength than BN fibers, while the thermal resistance and the dielectric properties of Si_3_N_4_ fibers are inferior to BN fibers [12]. Therefore, it is urgent to develop new types of ceramic fibers containing B, N, and Si to combine the merits of both BN and Si_3_N_4_ fibers in order to ensure excellent dielectric properties, high mechanical properties and good thermal resistance [13,14,15,16].

The polymer-derived ceramics (PDCs) method is the only feasible approach to prepare SiBN fibers; it can be used to design the atomic composition and the microstructure of fibers with low impurity [17,18,19,20]. Numerous researchers have synthesized different types of precursors for SiBN ceramic fibers. Tang et al. [21] obtained SiBN fibers using a novel precursor fabricated by the reaction of boron trichloride, dichloromethylsilane and hexamethyldisilazane and the obtained SiBN fibers had excellent mechanical strength up to 1.83 GPa as well as thermally stable dielectric properties. Liu et al. [22] prepared SiBN ternary ceramic fibers with a tensile strength of 0.87 GPa from a polymer precursor made from the reaction of hexamethyldisilazane, trichlorosilane, boron trichloride and methylamine. Peng et al. [23] prepared a precursor for SiBN ceramic fibers from the reaction of hexamethyldisilazane, silicon tetrachloride, boron trichloride and methylamine. In summary, a substantial amount of research has been focused on the fabrication of the spinnable polymer precursor containing the Si-N-B bridge bond, since the atomic ratio of Si:B:N of the preceramic polymer is difficult to adjust owing to its high activity [22]. In addition, the synthesis of single-source precursors usually needs multiple-stage processes, which results in significant yield loss and the unavoidable removal of by-products. Additionally, during the multi-step process, the polymeric intermediates are extremely sensitive to moisture, and the requirement for the large inert environment hampers its industrial applications [23].

In a previous study, Tan et al. [24] successfully synthesized BN/Si_3_N_4_ composite fibers using a mixture of polymers of poly(tri(methylamino)borizane) (PBN) and polysilazane (PSZ), and the fibers showed a mean tensile strength of 1040 MPa. Difficulties in the industrialization of this process exist mainly due to the fact that the PSZ is very sensitive to moisture. Polycarbosilane (PCS) is an organosilicon polymer with a normal structure—(CH_3_HSiCH_2_)_x_—that contains a –Si–C– backbone [25] and has been widely used as the precursor for preparation of silicon carbide and silicon nitride ceramic fibers owing to its good solubility and fusibility [26]. Therefore, we adopted a new strategy to prepare BN/Si_3_N_4_ ceramic fiber by using hybrid polymers of PBN and PCS. The proportions of BN and Si_3_N_4_ in the final ceramics could be easily adjusted through changing the ratios of PBN/PCS in hybrid composite polymers. The microstructural evolution of the obtained BN/Si_3_N_4_ composite ceramics and the properties of the composite fibers were investigated.

## 2. Experiments

### 2.1. Materials

PBN was synthesized by the reaction of BCl_3_ and NH_2_CH_3_, with a softening point of 80 ± 2 °C [27]. PCS (main units: (HSi(CH_3_)CH_2_), (CH_2_Si(CH_3_)_2_CH_2_), (Si(CH_3_)_2_Si(CH_3_)_2_)) with a number-average molecular weight of about 1150 and a softening point of 210 ± 2 °C was purchased from Zhongxing New Material Technology Co. Ltd., Ningbo, China. High purity N_2_ (>99.99%, Huanyujinghui Co. Ltd., Beijing, China) and NH_3_ (>99.99%, Nanfei Co. Ltd., Beijing, China) gases were used. All reactions were carried out under a dry nitrogen atmosphere.

### 2.2. Preparation of Composite Polymer

PBN and PCS were dissolved in toluene separately, with appropriate ratios, and mixed at 60 °C in a rotary evaporator (Shanghaiyukangkejiaoyiqishebei Co. Ltd., Shanghai, China) for 4 h to form homogeneous hybrid precursor solutions. Then, the solutions were dried at 80 °C for 2 h to remove the toluene. After cooling to room temperature, yellow transparent bulk solids were obtained. Six composite polymers (P1, P2, P3, P4, P5, P6) with different PBN/PCS ratios were prepared. After heating up to 1000 °C in ammonia at a heating rate of 1 K/min and 1600 °C in N_2_ at 2 K/min, the BN/Si_3_N_4_ composite ceramics with different Si_3_N_4_ mass contents were produced. The final BN and Si_3_N_4_ mass contents were calculated according to the ceramic yield of each polymer, which was 33 wt% and 58 wt% for PBN and PCS, respectively. The mass contents of PBN and PCS of each hybrid precursor (named P0–P7) as well as the BN and Si_3_N_4_ mass contents of the corresponding composite ceramics obtained at 1600 °C are listed in Table 1.

### 2.3. Preparation of Composite Fibers

Polymer green fibers were prepared using a lab-scale melt-spinning apparatus (Paigujingmijixie Co. Ltd., Beijing, China). The composite polymer was first fed into the spinning tube and heated to the spinning temperature (about 130 ℃) for 3 h. After the removal of the bubbles and an appropriate viscosity being obtained, the molten hybrid polymer was extruded through a single-capillary spinneret 0.20 mm in diameter by controlling the spinning pressure of N_2_ (roughly 0.5 MPa), and the unmelts were eliminated by a filter. Then, the extrudate flow was drawn into the filament uniaxially and collected on a spool with an appropriate rotating speed of 10 m/s.

Afterward, the green fibers were cured in ammonia at a heating rate of 0.1 K/min to 300 °C and pyrolyzed up to 1000 °C in flowing ammonia at a heating rate of 1 K/min. Then, the fibers were heated to 1600 °C under flowing N_2_ at 2 K/min. Finally, white composite ceramic fibers were obtained after cooling to ambient temperature.

### 2.4. Characterization

X-ray diffraction (XRD) studies were carried out with a PANalytical X’Pert-PRO diffractometer (Eindhoven, The Netherlands) at 2θ = 10–90° with Cu Kα radiation (λ = 0.15406 nm at 40 kV and 40 mA). The chemical bonding states were obtained by X-ray Photoelectron Spectroscopy (XPS, ESCALAB 250Xi) (Thermo Fisher Scientific, Waltham , Massachusetts, USA). The element analysis of silicon and boron was conducted using ICP-OES in a ThermoFisher iCAP6300 spectrometer (Waltham, MA, USA), and the nitrogen, carbon and oxygen contents were measured by a vario EL cube analyzer (elementar, Germany). Fiber morphologies were revealed by scanning electron microscopy (SEM) using a JSM-7001F system (JEOL, Tokyo, Japan). Element distribution along the fiber diameter was measured by an EPMA-1720 (Shimadzu, Japan). Single filament tensile properties were determined using an Instron5944 tensile tester (Norwood, MA, USA) with a gauge length of 25 mm, a load cell of 10 N, and a crosshead speed of 5 mm/min. The mean tensile strength of fibers was calculated based on 25 tested fibers using the Weibull statistic, and the Young’s modulus of the fibers was evaluated from the strain-stress curves. The dielectric properties of ceramic fibers determined at 10 GHz were measured by the short-circuited wave guide method using an Agilent HP8722ES vector network analyzer (Santa Clara, CA, USA) at ambient temperature based on the Chinese National Standard GB/T 5597-1999.

## 3. Results and Discussion

It is vital to study the high-temperature stability of polymer-derived ceramics. First, the microstructural developments of pure BN and Si_3_N_4_ pyrolyzed at different temperatures from PBN and PCS were investigated by XRD (Figure 1). For BN (Figure 1a), after being pyrolyzed at 1000 °C, two broad diffraction peaks at 2θ = 26.7° and 41.6° appeared. With the pyrolysis temperature rising from 1000 °C to 1600 °C, these diffraction peaks sharpened a little because of ongoing BN crystallization. At 1600 °C, the diffraction peaks were still broad, and no resolutions of the (100) or (101) doublet were displayed, which indicated the formation of BN nanocrystallines. For Si_3_N_4_ (Figure 1b), it can be seen that the as-pyrolyzed ceramics were amorphous below 1400 °C, and the crystallization process started at 1500 °C, which would affect the high temperature stability of ceramics or ceramic fibers.

To elucidate microstructural evolution of these composite polymer-derived ceramics, six hybrid precursors with different PBN/PCS ratios were fabricated and pyrolyzed at 1600 °C. The calculative mass contents of BN and Si_3_N_4_ in the composite ceramics (P0-7-C) are listed in Table 1. The chemical environment of B, N and Si atoms in the composite ceramic (P3-C) was studied by XPS (Figure 2). The B_1s_ peak at 190.8 eV and the N_1s_ at 398.1 eV confirmed the presence of BN. Moreover, the binding energy centered at 102.5 eV for Si_2p_ and 399.7 eV for N_1s_ demonstrated the existence of Si–N bonds, indicating that the composite ceramic was composed of a mixture of BN and Si_3_N_4_.

The elemental compositions of these polymer-derived composite ceramics obtained at 1600 °C (P0–7-C) are listed in Table 2. Additionally, XRD patterns of these composite ceramics are shown in Figure 3. Obviously, when the content of the BN was over 75 wt% in the composite ceramics, no silicon nitride peaks were detected, indicating that the BN phase restrained the decomposition of Si_3_N_4_ and limited the grain size of Si_3_N_4_ crystals. Likewise, with the increase of Si_3_N_4_ mass content in these composite ceramics, the crystallization process of BN was also hindered, and there existed no BN peaks when the mass content of Si_3_N_4_ exceeded 15 wt%. Noticeably, the P3-C containing 75 wt% BN and 25 wt% Si_3_N_4_ was totally amorphous, which indicated that the clusters of BN and Si_3_N_4_ totally hindered the crystallization of each other. Its amorphous state at 1600 °C would guarantee reliable performance for the composite ceramics or ceramic fibers in high-temperature environments. However, unlike the results of Tan et al. [14], who used PSZ as the raw materials rather than PCS, no h-BN crystals were found in this system. After analyzing the oxygen contents of PSZ and PCS, which were 3% and 0.6%, respectively, it was concluded that the introduction of oxygen could promote the crystallization process of h-BN by formulating the oxides with low melting points.

The composite ceramics pyrolyzed at 1600 °C were further annealed at 1700 °C under N_2_ for 2 h, and the corresponding XRD patterns are shown in Figure 4. Except for P1-C, the composite ceramics all exhibited Si_3_N_4_ crystals, which was not beneficial for the high-temperature stability of the composite ceramics. In summary, the BN/Si_3_N_4_ composite ceramics could only keep stability below 1700 °C with appropriate ratios.

Based on the investigation of the microstructural evolution of BN/Si_3_N_4_ composite ceramics, the crystallization of Si_3_N_4_ could be totally restrained when its content in the composite ceramics was below 25% at 1600 °C. Then, the composite ceramic fibers were fabricated from hybrid precursor P1, P2 and P3 through melt-spinning, curing and decarburization in NH_3_ under 1000 °C and pyrolysis at 1600 °C in N_2_. The elemental compositions of these obtained fibers (P1-F, P2-F and P3-F) are listed in Table 3. Additionally, the XRD spectra of these fibers are shown in Figure 5. All these fibers only showed two broad diffuse peaks, revealing the low crystallinity of BN [28,29]. Noticeably, no diffraction peaks of Si_3_N_4_ in the composite fibers were detected, indicating that Si_3_N_4_ existed in an amorphous state, which was beneficial to the high temperature stability of the composite fibers.

Figure 6 shows the morphologies of the obtained BN/Si_3_N_4_ composite fibers pyrolyzed at 1600 °C. The diameter of the fibers was roughly 12 μm, and the surface was smooth and compact, without any apparent voids. The cross sections were nearly circular without inter-fusion, exhibiting a glass-like fracture feature, which demonstrated that the curing and pyrolysis process could meet the preparation requirements.

In order to clarify the distributions of these elements (Si, B, N), the fibers were embedded in epoxy resin, with further polishing and spraying carbon, and then characterized by EPMA (Figure 7). Obviously, the distributions of each atom for P1-F and P2-F were nearly homogeneous. For P3-F, the atomic Si aggregated in the core, while the concentration of atomic B was higher in the outside shell, revealing the phase separation of Si_3_N_4_ and BN. It was concluded that during the spinning process, the PCS tended to aggregate in the core under the shearing pressure owing to the huge differences of the viscosity and softening point of PBN (80 °C) and PCS (210 °C), which led to the unique structure of the final composite fiber; this structure could only be obtained when the Si_3_N_4_ content of the fibers reached 25 wt%.

The Weibull plots of failure strength of these fibers are illustrated in Figure 8. The tensile strength, Young’s modulus, Weibull modulus and dielectric properties (f = 10 GHz) of BN/Si_3_N_4_ fibers are listed in Table 4. The tensile strength of BN/Si_3_N_4_ fibers rose dramatically with the increase of Si_3_N_4_ mass content, and that of P3-F reached 1360 MPa with the Young’s modulus of 117 GPa. Compared with the results of Tan et al. [24], the composite fiber (P3-F) showed no h-BN crystals, but a higher tensile strength. Except in the case of the slightly higher mass content of Si_3_N_4_ in P3-F, the micro-cracks were caused by the crystallization process of h-BN. Therefore, in order to enhance the tensile strength of BN/Si_3_N_4_ composite fibers, the crystallization process of h-BN crystals should be avoided and the mass content of Si_3_N_4_ should be enhanced as much as possible in an appropriate ratio range of BN/Si_3_N_4_, where Si_3_N_4_ could remain amorphous in the final composite fibers. Apart from the excellent tensile strength, the composite fibers (P3-F) showed a low dielectric constant of 3.34 and loss tangent of 0.0047 at 10 GHz. The excellent dielectric properties could be ascribed to the low carbon content [30], which was less than 0.1 wt%. The combination of improved mechanical properties and excellent dielectric behavior demonstrated the potential for wave-transparent applications.

## 4. Conclusions

BN/Si_3_N_4_ composite ceramics and ceramic fibers were obtained through the precursor-derived ceramic route using the hybrid polymers of poly(tri(methylamino)borazine) (PBN) and polycarbosilane (PCS). When the Si_3_N_4_ content of the composite ceramics was below 25 wt%, no Si_3_N_4_ crystals were found at 1600 °C, and the BN/Si_3_N_4_ composite ceramic containing 25 wt% Si_3_N_4_ was totally amorphous. Three kinds of BN/Si_3_N_4_ composite fibers containing 5 wt%, 15 wt% and 25 wt% Si_3_N_4_ were fabricated successfully, showing the nanocrystallines of BN and amorphous Si_3_N_4_, of which the mean tensile strength and Young’s modulus was enhanced with the increasing of the Si_3_N_4_ mass content. Additionally, the composite fibers (P3-F) showed a unique BN (shell)/Si_3_N_4_ (core) structure, with average tensile strength of 1.36 GPa and Young’s modulus up to 117 GPa. Moreover, the composite fiber (P3-F) exhibited excellent dielectric properties, with a dielectric constant of 3.34 and a dielectric loss tangent of 0.0047 at 10 GHz. Further research is in progress to optimize the oxygen content to improve the mechanical properties of the composite ceramic fibers.

## Figures and Tables

**Figure 1 materials-14-06194-f001:**
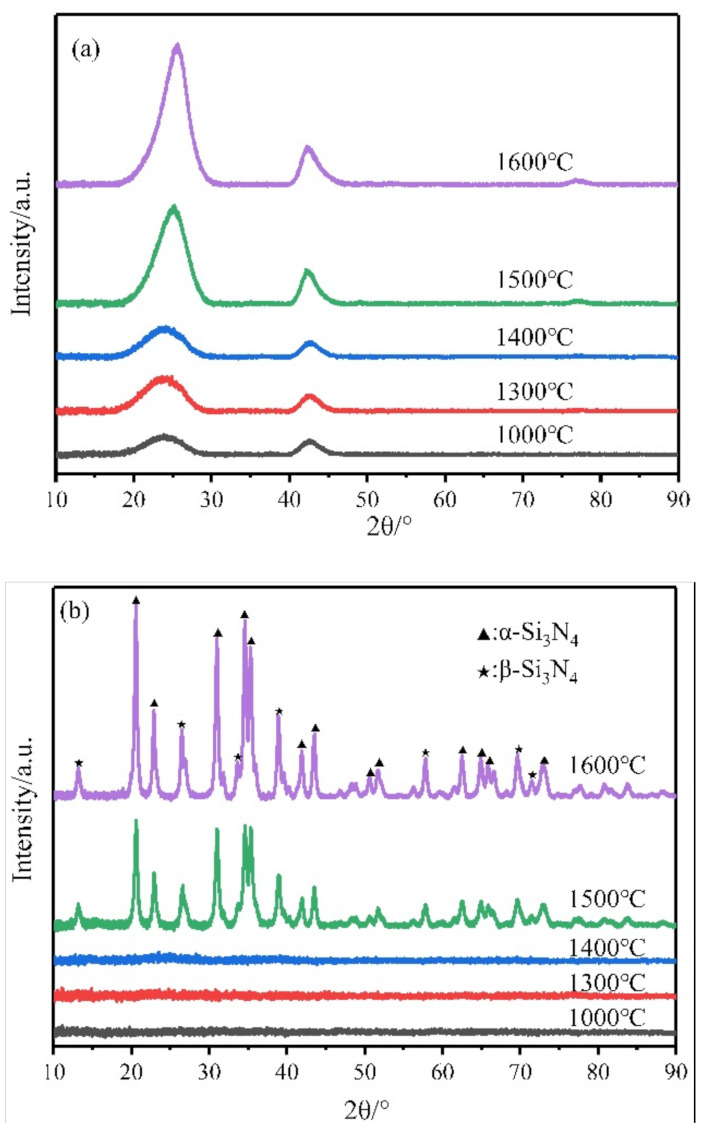
XRD patterns of (**a**) BN and (**b**) Si_3_N_4_ pyrolyzed at different temperatures from PBN and PCS.

**Figure 2 materials-14-06194-f002:**
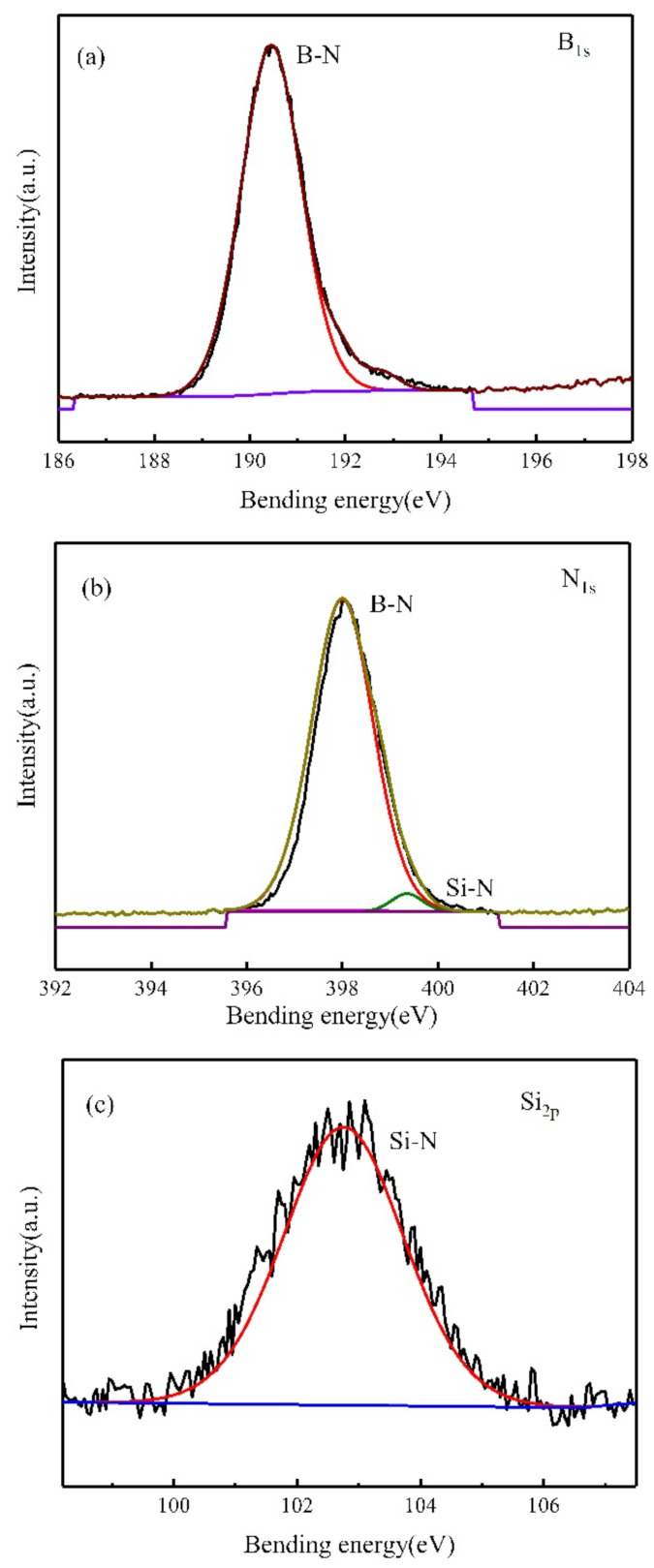
XPS spectra of the composite ceramic (P3-C): (**a**) B_1s_; (**b**) N_1s_; (**c**) Si_2p_.

**Figure 3 materials-14-06194-f003:**
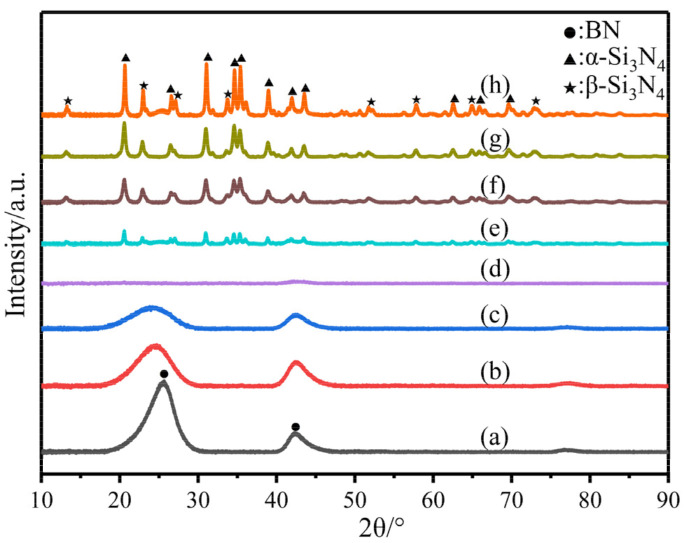
XRD patterns of the composite ceramics pyrolyzed at 1600 °C: (**a**) P0-C; (**b**) P1-C; (**c**) P2-C; (**d**) P3-C; (**e**) P4-C; (**f**) P5-C; (**g**) P6-C; (**h**) P7-C.

**Figure 4 materials-14-06194-f004:**
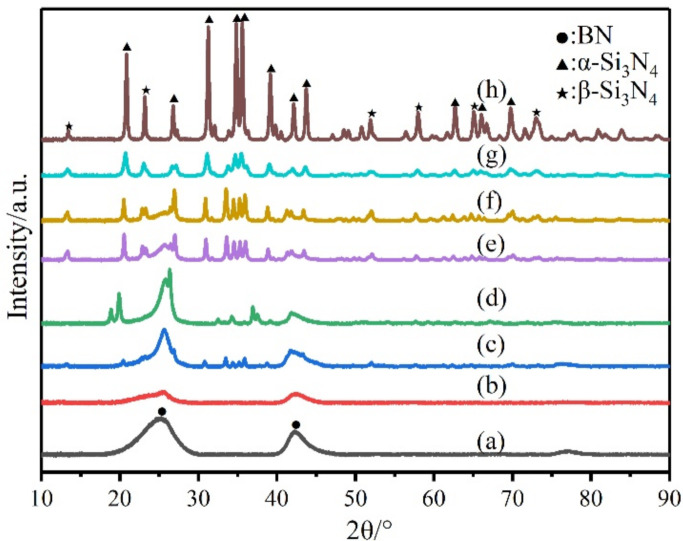
XRD patterns of the composite ceramics pyrolyzed at 1700 °C: (**a**) P0-C; (**b**) P1-C; (**c**) P2-C; (**d**) P3-C; (**e**) P4-C; (**f**) P5-C; (**g**) P6-C; (**h**) P7-C.

**Figure 5 materials-14-06194-f005:**
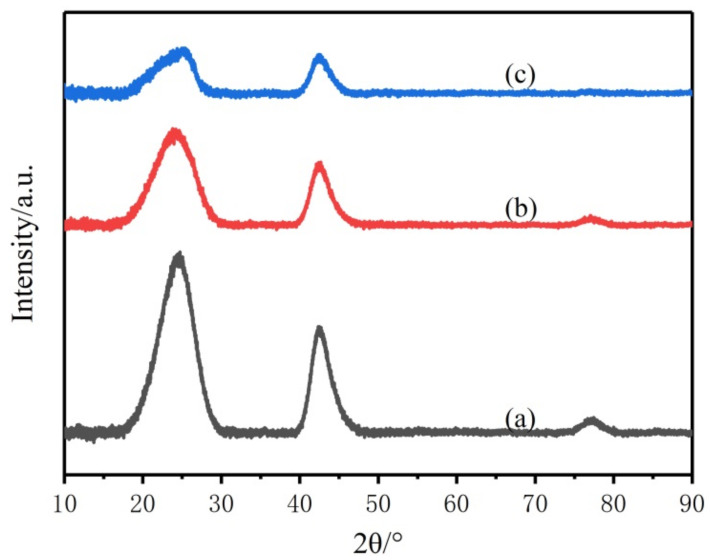
XRD patterns of the composite ceramic fibers: (**a**) P1-F; (**b**) P2-F; (**c**) P3-F.

**Figure 6 materials-14-06194-f006:**
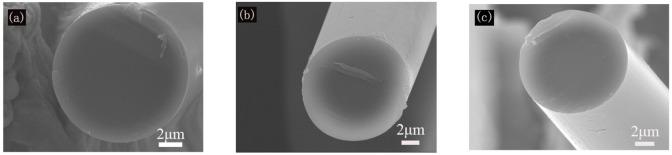
SEM images of the surface and cross sections of the composite fiber: (**a**,**d**) P1-F; (**b**,**e**) P2-F; (**c**,**f**) P3-F.

**Figure 7 materials-14-06194-f007:**
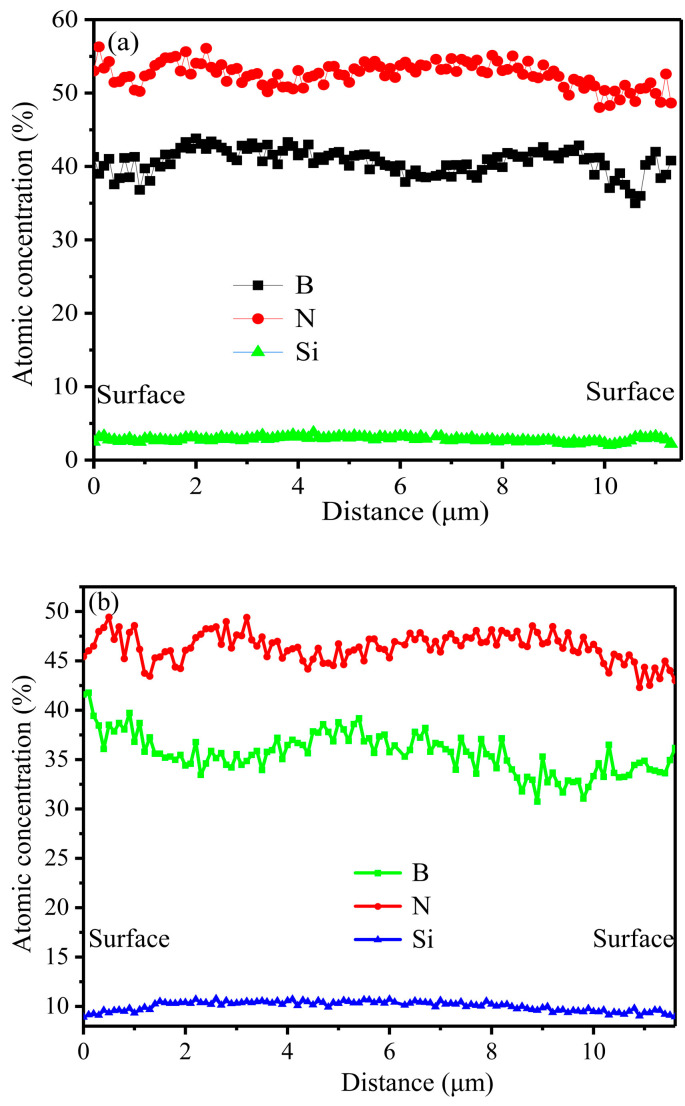
Elemental concentration of B, N and Si along the composite fiber diameter: (**a**) P1-F; (**b**) P2-F; (**c**) P3-F.

**Figure 8 materials-14-06194-f008:**
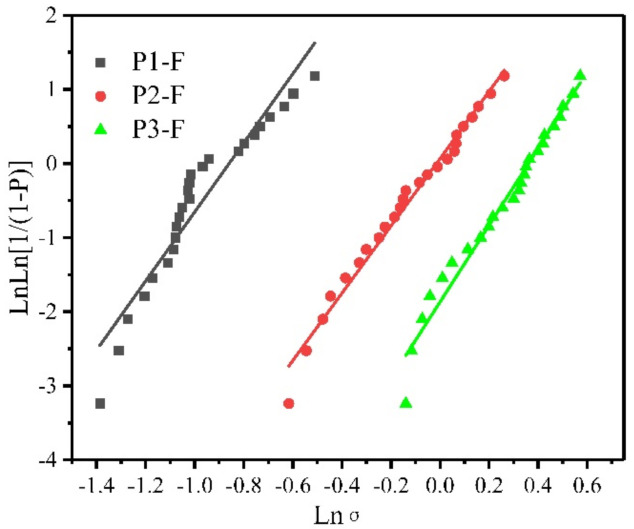
Weibull plot of failure strengths of three composite fiber diameters.

**Table 1 materials-14-06194-t001:** Composition of hybrid PBN/PCS precursors and the BN/Si_3_N_4_ ceramics derivatives (1600 °C).

Materials	Precursor Composition (wt%)		Ceramics Composition (wt%)	
	PBN	PCS	BN	Si_3_N_4_
P0	100	0	100	0
P1	97.2	2.8	95	5
P2	91.1	8.9	85	15
P3	84.4	15.6	75	25
P4	77.0	23.0	65	35
P5	64.3	35.7	50	50
P6	37.5	62.5	25	75
P7	0	100	0	100

**Table 2 materials-14-06194-t002:** Elemental content of different BN/Si_3_N_4_ composite ceramics.

Elemental Content (wt%)	Si	B	N	C
P0-C	0	43.7	56.1	0.2
P1-C	3.4	41.3	55.2	0.1
P2-C	8.7	36.9	54.2	0.2
P3-C	15.6	32.6	51.7	0.1
P4-C	19.7	29.3	50.8	0.2
P5-C	29.5	21.7	48.7	0.1
P6-C	43.2	10.4	44.3	0.3
P7-C	60.1	0	39.8	0.1

**Table 3 materials-14-06194-t003:** Elemental content of BN/Si_3_N_4_ composite fibers.

Elemental Content (wt%)	Si	B	N	C
P1-F	3.2	41.3	55.3	0.2
P2-F	8.2	37.2	54.5	0.1
P3-F	14.6	33.1	52.3	0.1

**Table 4 materials-14-06194-t004:** Tensile strength, Young’s modulus and dielectric properties (f = 10 GHz) of BN/Si_3_N_4_ composite fibers.

Material	Tensile Strength (MPa)	Young’s Modulus (GPa)	Weibull Modulus	Dielectric Constant	Loss Tangent
P1-F	365	35	4.65	3.02	0.0023
P2-F	832	93	4.53	3.21	0.0031
P3-F	1360	117	5.17	3.34	0.0047

## Data Availability

Not applicable.

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
