# Peer review of "Characterization and Microstructural Evolution of Continuous BN Ceramic Fibers Containing Amorphous Silicon Nitride"

_materials, 2021, doi:10.3390/ma14206194_

Round 1

Reviewer 1 Report

The author reported the BN/Si3N4 composite fiber with excellent mechanical properties. This manuscript is publishable after revision.

There are several works on BN ceramic fibers containing silicon nitride.  In this study, they reported high tensile strength, which is important in many fields, however, the author should explain more in the manuscript for high tensile strength.  

In XRD, the silicon nitride peaks should be labeled properly. The peaks contain both α and β silicon nitride phases.

How do author knows the silicon nitride is in amorphous phase? The author reported the core-shell structure, the silicon nitride in core may be crystalline. I suggest author to get diffraction pattern of core by TEM (SAED).

Need some English correction.

Reviewer 2 Report

The manuscript "Characterization and microstructural evolution of continuous BN ceramic fibers containing amorphous silicon nitride" is reviewed. The work is novel and results are presented very well. This is a suitable article for publication in Materials. Only some minor corrections are required. 

1- There are some grammatical and typos mistakes:

  • Line 15: "amount"
  • Line 59: "Therefore, we developed a new strategy in this article for a novel"
  • Line 65: What do you mean by "The systematic studies on the microstructure and properties of the composite fiber will be presented."
  • Line 83: "content were"

2- Why you did not compare or report fibers such as SiC?

3- Why you did not use FTIR for investigating the chemical interactions?

4- What is the hardness of the synthesized ceramic? 

5- You mentioned "optimizing the oxygen content to improve the mechanical properties of the composite ceramic fibers.". Please explain how?

Round 2

Reviewer 1 Report

The author revised the manuscript  according to reviewer's comments and suggestions. Therefore, the revised version should be accepted.